# Assessment of Hydrologic Alteration Metrics for Detecting Urbanization Impacts

**Reid D. McDaniel and Frances C. O'Donnell \***

Department of Civil Engineering, Auburn University, 238 Harbert Engineering Center, Auburn, AL 36849, USA; rdm0015@auburn.edu
\* Correspondence: fco0002@auburn.edu; Tel.: +1-334-844-7168

**Abstract:** Urbanization is increasing rapidly and has the potential to alter the hydrologic cycle. It is uncertain if hydrologic alteration metrics developed for large-scale analyses detect the impacts of urbanization. This study tests the ability of two such methods, Indicators of Hydrologic Alteration (IHA) and streamflow signatures, to detect the effects of urbanization in two watersheds in the southeastern U.S.A. A hydrologic model (HEC-HMS) was used to simulate flows in ungauged upstream tributaries to determine if analysis of flow from a large gauged watershed detects urbanization effects on upstream tributaries. IHA analysis detected trends in time in the watersheds, but the results were the opposite of what would be expected as urbanization increased minimum flows, decreased maximum flows, and decreased flashiness based on the trend in time and comparison with an undeveloped watershed. IHA parameters were more sensitive to urbanization than streamflow signatures. Subcatchments that transitioned from low to moderate or high levels of urbanization had greater levels of hydrologic alteration than was detected at the watershed outlet. Analyses of stream gauge network data may underestimate the importance of urbanization as a watershed characteristic due to scale issues, the variable effects of water management, and the dynamic nature of urbanization.

**Keywords:** streamflow; indicators of hydrologic alteration; streamflow signatures; urbanization; water management; scale issues; watershed modeling

## 1. Introduction

Urbanization is a rapidly increasing phenomenon. By 2030, it is expected that 84% of the U.S. population will reside in urban areas [1], with urban land cover expanding to three times its area in 2000 [2]. The increase in impervious area associated with urbanization alters the hydrologic cycle by decreasing infiltration and evapotranspiration and increasing surface runoff [3,4]. This can have negative impacts for urban communities and those downstream, such as flooding and reduced water quality [5]. Urbanization also impacts aquatic and riparian habitat by changing the flood regime, altering channel geomorphology, increasing the dominance of tolerant species, and reducing biotic richness [6,7].

Many of the impacts of urbanization on hydrologic dynamics involve alteration of the hydrograph, the timeseries of streamflow observed in response to precipitation in the watershed [8]. Peak flows are expected to increase, and recession times are expected to decrease due to the reduction in travel time of water over impervious surface versus undeveloped land [9]. The low flows of some streams have also been shown to decrease after urbanization due to the reduction in subsurface flows [10], though the effect of urbanization on baseflow is complex and may lead to increases or decreases [11]. This overall suite of characteristics, along with alterations to water quality and ecological health, have been described as urban stream syndrome [12]. More recent analyses of urban stream syndrome

have found that the response of streams to urbanization is heterogeneous with variation due to both natural and human factors [13].

Increasingly, hydrologists are using "big data" approaches to reveal patterns, associations, and trends, especially relating to human impacts on and interactions with hydrologic systems [14]. Networks of long-term streamflow records, such as the U.S. Geological Survey (USGS) National Water Information System (NWIS), are a key dataset for characterizing hydrograph response in many such analyses (e.g., [15,16]). Numerous studies have used machine learning and data mining analysis of streamflow characteristics to identify hydrologically similar watersheds and relate these groupings to watershed characteristics [17–19]. However, there are several reasons that a regional-scale or larger study of stream gauge network data may fail to detect the impact of urbanization on hydrology and thus underestimate the importance of urbanization as a driver of watershed dynamics.

First, stormwater management, which seeks to reduce the impact of urbanization on hydrology, may mask the effects of urbanization observed at a stream gauge to varying degrees [20]. In the U.S.A., federal laws limit the discharge of pollutants in urban runoff from point and nonpoint sources [21]. However, water quantity regulations are mandated by local or state governments and vary by location. Most water quantity regulations specify that post-development peak flows must be no higher than pre-development peak flows, but the required design storm or storms vary by location [22]. Some states or municipalities have requirements other than peak flow reduction, such as requirements for maintaining groundwater recharge within a developed area (e.g., [23]). These design requirements, as well as available technology, prevalent practices, and infrastructure age, will change over time [20,24]. For example, Loperfido et al. [25] found that distributed stormwater management, a practice that has become more common recently, resulted in greater baseflow and lower peak flows than centralized stormwater management.

Second, many gauged watersheds with long-term data are quite large [26], and urbanization is unlikely to occur uniformly across a watershed [27]. Studies of urbanization on small catchments ($<25$ km$^2$) consistently show higher peak flows and a flashier hydrograph [5,28], but studies on larger watersheds (approximately 50–1000 km$^2$) within the same region show inconsistent responses to urbanization [4,29,30]. The goal of many analyses of urbanization effects on hydrology is to characterize the potential of urbanization to impact aquatic habitat. Decreases in the size and diversity of fish populations in a stream are closely related to the percent of impervious area in its catchment [7,10], but this is scale dependent. A heavily urbanized small headwater catchment could experience substantial impacts even if the percent impervious area in the watershed as a whole is low. Additionally, studies conducted across many watersheds must use streamflow parameters that do not scale with catchment size [18], and it is uncertain if these parameters capture urbanization impacts effectively.

Finally, urbanization is not a static characteristic of a watershed, like slope or soil type. Urbanized area has expanded globally over recent decades, which covers the period of record for most stream gauging networks. Among the regions of the U.S., urbanization has expanded most rapidly in the Southeast since 1980 [31,32], and several analyses of urbanization impacts on hydrology have been performed in this region [4,5,8,28]. Whether urbanization is treated as a static or dynamic watershed characteristic influences the results. Rose and Peters [8] used a comparative approach of analyzing urbanized watersheds and nearby watersheds with minimal urban land cover, which treats urbanization as a static characteristic. The study concluded that urbanization does not impact runoff ratio or the amount of precipitation converted to runoff. A later study by Diem et al. [4] was conducted in the same region with some of the same watersheds. It treated urbanization as a dynamic process by analyzing the trend in streamflow statistics from 1982–2015 for watersheds where urban area expanded. Significant increases in runoff were observed, suggesting urbanization does increase runoff ratio. However, other factors that could drive trends in streamflow, such as decadal-scale climate variability and groundwater pumping [33] could be confounded with urbanization in studies conducted in this manner. Within watersheds that are already heavily urbanized, aging or replacement

of water infrastructure may lead to trends in streamflow characteristics [20]. For example, Peachtree Creek in Atlanta, Georgia, U.S., a major city, had a significant declining trend in streamflow over three decades [4], which may be due to increasing leakiness of storm sewers or retrofitting with low impact development practices that promote infiltration.

This study tests the ability of two large-scale analysis methods, streamflow signatures and Indicators of Hydrologic Alteration (IHA) to detect the impact of urbanization in two watersheds in the southeastern U.S. that have urbanized rapidly over the past three decades. Streamflow signatures have been used to investigate the role of climate, land-use, and physiographic properties in determining stream flow regime [34]; to optimize hydrometric network design [35]; and to perform supervised (e.g., [36]) and unsupervised (e.g., [15]) classification of watersheds. IHA analysis has been used to study urbanization effects on streamflow at a range of spatial scales [5,37]. A before-after control-impact (BACI) approach [38] is used in which the watersheds are analyzed both for temporal trends and in comparison to a nearby minimally developed watershed with similar characteristics. A hydrologic model is used to determine if analysis of flow from a large gauged watershed detects urbanization effects on upstream tributaries. Three research questions are addressed:

1. Do streamflow signatures and IHA analysis detect urbanization through either comparison of pre- and post-urbanization data or comparison with a minimally developed watershed?
2. Is the change detected consistent across watersheds within a region and with the expected increase in flashiness associated with urbanization?
3. Does analysis of stream gauge network data detect urbanization in upstream tributaries?

## 2. Materials and Methods

### 2.1. Study Sites

Two pairs of watersheds in the southeastern U.S. were selected for the study, with each pair composed of one "urbanized" watershed (U1 and U2) and one "comparison" watershed (C1 and C2). All watersheds in the states of Alabama and Georgia with streamflow data in the USGS NWIS database were considered for analysis. Adequate sites for analysis had at least 30 water years of complete daily data and had precipitation data available from a gauge in or near the watershed. The two urbanized watersheds were selected from these sites because they had an increase in urban land cover between 1992 and 2011 and had a nearby watershed of similar size and soil type with little to no change in urban land cover for comparison. The first pair of watersheds, U1 and C1, is located near Birmingham, Alabama, and the second, U2 and C2, is located near Atlanta, Georgia (Figure 1). Watershed U2 is somewhat larger than the other watersheds. Therefore, comparisons of the magnitude of parameters that are not normalized by area should be done cautiously when this watershed is considered. C2, despite being smaller, still offers a useful comparison for trends in parameters and magnitude of normalized parameters such as streamflow signatures. The climate in the region is humid subtropical with mild winters and hot, humid summers. All watersheds are on sandy loam soil. Additional watershed characteristics are given in Table 1.

**Table 1.** Study watershed characteristics including the location of the outlet gauging station and the year that was determined as the breakpoint between the before and after urbanization periods.

| Watershed | Drainage Area (km²) | Mean Annual Precip. (mm) | Mean Annual Temp. (°C) | Outlet Location Latitude | Longitude | Breakpoint Year |
|---|---|---|---|---|---|---|
| U1 (Valley Creek) | 383 | 1437 | 17.3 | 33.44752 | −87.12190 | 1996 |
| C1 (Big Canoe Creek) | 365 | 1403 | 16.3 | 33.84005 | −86.26276 | 1996 |
| U2 (Sweetwater Creek) | 616 | 1282 | 16.0 | 33.77296 | −84.61415 | 1996 |
| C2 (New River) | 329 | 1279 | 16.3 | 33.23526 | −84.98775 | 1996 |

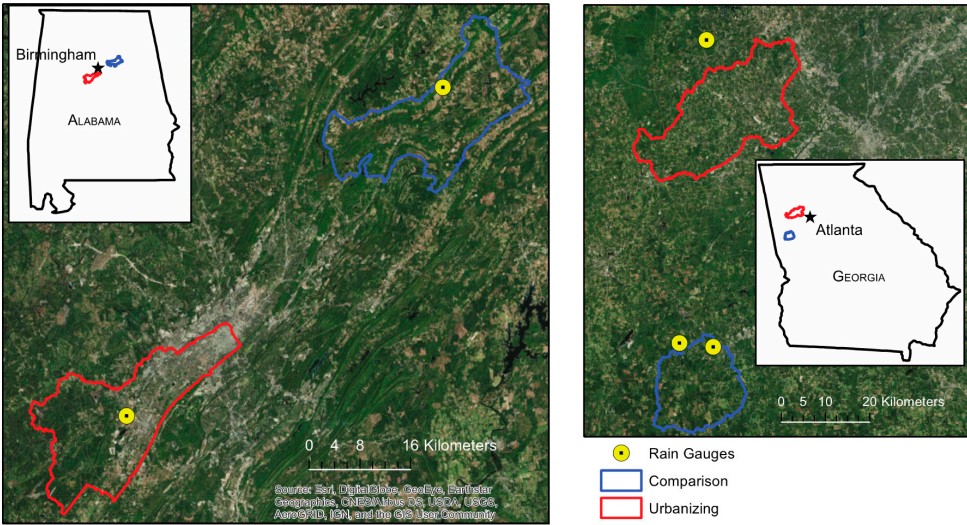

**Figure 1.** Location of study watersheds and rain gauges in Alabama and Georgia, USA.

## 2.2. Data Sources

Streamflow data were downloaded from the NWIS (https://waterdata.usgs.gov/nwis) for sites 02462000 Valley Creek near Oak Grove AL (U1), 02401390 Big Canoe Creek at Asheville AL (C1), 02337000 Sweetwater Creek near Austell GA (U2), and 02338660 New River at GA 100 near Corinth GA (C2). Land cover change data were obtained from the National Land Cover Database (NLCD), which are available through the Multi-Resolution Land Characteristics Consortium (https://www.mrlc.gov/), for the time periods of 1992–2001, 2001–2006, and 2006–2011. NLCD raster files were clipped to the watershed boundaries, and the number of pixels that were reclassified as urbanized land cover types in each time period was determined to quantify urbanization. Within the time period when most urbanization occurred, Google Earth imagery (https://www.google.com/earth/) was visually inspected for each urbanized watershed to select a year when the most change occurred. This year served as the breakpoint between before and after urbanization time periods in subsequent analyses.

Daily precipitation data were obtained from the National Weather Service Database (https://xmacis.rcc-acis.org/). Stations used were Bessemer 3 WSW for U1, Springville 6.5 NE for C1, Kennesaw 3.1 SSW for U2, and Newnan 7 WNW and Newnan 2.6 W for C2. All stations were within the watershed boundaries, except for Newnan 7 WNW, which is within 5 km of the watershed boundary. Daily data from the Parameter-elevation Regressions on Independent Slopes Model (PRISM; [39]) were used gap fill missing gauge data. For hydrologic modeling, a 1/3 arcsecond (approximately 10 m) Digital Elevation Model (DEM) and Hydrologic Unit Code (HUC)-8 subbasin hydrography dataset were obtained for the study area from the USGS National Map Viewer [40]. A map of hydrologic soil group classifications from the Soil Survey Geographic Database (SSURGO) was obtained from the Natural Resource Conservation Service (NRCS) Web Soil Survey [41].

## 2.3. Indicators of Hydrologic Alteration

The watersheds were analyzed for hydrologic alteration over time using the IHA application developed by The Nature Conservancy [42]. The software uses a before and after approach to determine changes to flow frequency, magnitude, and duration. Mean daily streamflow over the period of record is the input to the application. The parametric analysis option was used to allow for the comparison of means and standard deviations of hydrologic indicators across watersheds and analysis periods. IHA calculates 32 ecologically relevant hydrologic alteration parameters. IHA was first run using the single period option to determine if there was a statistically significant trend over time in each IHA parameter. The software performs a linear regression analysis for each parameter with year as the independent variable over the period of record and calculates a *p*-value for the regression line to

indicate the significance of the trend. Any parameters with a significant trend ($p < 0.05$) were included in a two-period analysis that compared the before and after urbanization periods. The mean and coefficient of variation were calculated for the parameters in each period to determine the change in center and dispersion between the two periods. Data availability allowed for the IHA analysis to be performed for water years 1979 through 2017, which is long enough to account for decadal-scale climate variability [4,8,43].

The IHA application characterizes the change in parameters using a range of variability approach (RVA) [44]. For each hydrologic alteration parameter, three categories are established using the mean and standard deviation for the before urbanization period: (1) *low* for values more than one standard deviation below the mean, (2) *middle* for values within one standard deviation of the mean, and (3) *high* for values more than one standard deviation above the mean. A hydrologic alteration (*HA*) value is then calculated for each parameter and each category as

$$HA = \frac{N_A - N_B}{N_B},\tag{1}$$

where $N_A$ and $N_B$ are the number of water years with values in the category in the after and before urbanization periods, respectively. Minimal hydrologic alteration is indicated by values close to zero, whereas a positive value indicates that the frequency of values in the category increased and negative values indicate that the frequency decreased.

*2.4. Streamflow Signature Analysis*

Streamflow signatures were determined from streamflow and precipitation data through analysis in Matlab 2016b (Mathworks, Natick, MA, USA). The signatures do not scale with catchment size, are not limited by calibration or model errors, and have an interpretable link to watershed characteristics [45]. Four streamflow signatures were calculated: runoff ratio, slope of the flow duration curve (FDC), streamflow elasticity, and baseflow index. Streamflow signature analysis requires both precipitation and streamflow data. Due to limitations in precipitation data availability, the analysis was only performed for water years 1982 to 2017. Runoff ratio ($R_{QP}$) is the proportion of precipitation that becomes streamflow at the annual scale, and is given by

$$R_{QP} = \frac{Q}{P},\tag{2}$$

where $Q$ and $P$ are the sums of streamflow and precipitation for a water year. The FDC is the cumulative distribution function of daily streamflow subtracted from one giving the probability that a streamflow value will be equaled or exceeded at a randomly selected time. The slope of the FDC ($S_{FDC}$) is given by

$$S_{FDC} = \frac{\ln(Q_{33\%}) - \ln(Q_{66\%})}{(0.66 - 0.33)},\tag{3}$$

where $Q_{33\%}$ is the 33rd percentile streamflow and $Q_{66\%}$ is the 66th percentile streamflow [46]. It is an indicator of the 'flashiness' of a watershed. High magnitude values indicate frequent high and low flows, while low magnitude values indicate a more damped flow regime. Streamflow elasticity ($E_{QP}$) indicates a watershed's sensitivity to precipitation changes. This is defined as the proportional change in streamflow divided by the proportional change in precipitation [47] and is given by

$$E_{QP} = \text{median}\left(\frac{dQ}{dP}\frac{P}{Q}\right),\tag{4}$$

where $dQ$ and $dP$ are the difference between the previous year and current year streamflow and precipitation, respectively. The baseflow index (*BFI*) gives the proportion of streamflow that is baseflow and is given by

$$BFI = \frac{Q_B}{Q},$$ (5)

where $Q_B$ is the mean baseflow. Baseflow was determined using the one parameter single pass digital filter method [48] with a daily time step. Baseflow for a time step ($Q_{Bt}$) is the difference between total streamflow for the time step ($Q_t$) and direct flow for the time step ($Q_{Dt}$), given by

$$Q_{Dt} = cQ_{Dt-1} + \frac{1+c}{2}(Q_t - Q_{t-1}),$$ (6)

where $Q_{Dt-1}$ and $Q_{t-1}$ are the direct flow and total flow in the previous time step, respectively, and $c$ is a constant set equal to 0.925 [49].

The streamflow signatures were calculated for each water year from 1982 to 2017. To determine if there was a significant trend in streamflow signatures over this period, a least-squares linear regression line was fit to the data and an *F* test was used to determine if the slope was significantly different from zero indicating a significant trend.

## 2.5. Hydrologic Modeling

A hydrologic model was created for watershed U1 using the Hydrologic Engineering Center Hydrologic Modeling System (HEC-HMS) 4.2.1 [50]. It was used to simulate streamflow at ungauged locations in upstream tributaries to determine if IHA and streamflow signature analysis of data for the stream gauging network were able to detect the impact of urbanization on upstream catchments. Two model scenarios were developed, representing the before and after urbanization periods. The hydrologic model was created from elevation and hydrography datasets using the ArcHydro and GeoHMS toolboxes in ArcMap 10.3.1 (ESRI, Redlands, CA, USA). Terrain preprocessing was performed using a stream threshold of 20,000 elevation raster cells (approximately 2 km²) to allow for delineation and analysis of small upstream tributaries. A Soil Conservation Service (SCS) curve number grid was created from NLCD land use data, SSURGO hydrologic soil groups, and the SCS TR-55 manual [51] as described in Appendix A.

The SCS method was used for the loss and transform methods. The recession method was used for baseflow, which uses standard baseflow separation techniques and exponentially declining baseflow that represents drainage from storage in the watershed. The Muskingum method was used for river routing. The Muskingum *K* and *X* parameters, initial baseflow, baseflow recession constant, baseflow ratio to peak, and initial abstraction were determined through model calibration and validation as described in Appendix A. Each subbasin was paired with the nearest rain gage, and daily precipitation data were input as a specified hyetograph.

Major subcatchments were identified and land cover change was analyzed by subcatchment using the methods described in Section 2.2 to determine if urbanization rates were heterogeneous within the watershed. The HEC-HMS model was run for the study period of 1982 to 2017. IHA and streamflow signature analyses were performed on the modeled flows at subcatchment outlets as described in Sections 2.3 and 2.4, respectively.

## 3. Results

### 3.1. Land Cover Change

Most land cover change over the study period was the conversion of area classified as forest or as pasture and hay to suburban/residential or medium intensity commercial classifications, representing urbanization. The NLCD 1992–2001 time period had the greatest change in urban area (Table 2). Urbanized area increased by 35.8 km² in watershed U1 and 31.6 km² in watershed U2. Between 2001 and 2006, 34.5 km² and 30.4 km² was urbanized in U1 and U2, respectively. Between 2006 and 2011, 16.3 km² and 10.4 km² was urbanized in U1 and U2, respectively. Less than 3 km² was urbanized

between 1992 and 2011 in both comparison watersheds. Qualitative inspection of Google Earth imagery identified water year 1996 as the start of the after-urbanization period for both watersheds.

**Table 2.** Increase in urbanized area in each watershed over the three time periods covered by National Land Cover Database (NLCD) data.

| Watershed | 1992–2001 | | 2001–2006 | | 2006–2011 | |
|:---:|:---:|:---:|:---:|:---:|:---:|:---:|
| | Area (km$^2$) | % of Area | Area (km$^2$) | % of Area | Area (km$^2$) | % of Area |
| U1 | 35.8 | 9.3% | 34.5 | 9.0% | 16.3 | 4.3% |
| C1 | 1.0 | 0.3% | 1.3 | 0.4% | 0.3 | 0.1% |
| U2 | 31.6 | 5.1% | 30.4 | 4.9% | 10.4 | 1.7% |
| C2 | 1.2 | 0.4% | 1.6 | 0.5% | 0.2 | 0.1% |

*3.2. IHA Analysis*

The single period trend analysis indicated a significant trend for many hydrologic alteration parameters for watersheds U1 and C2. Trends were evident in U1 for low flow and high flow parameters (1-, 3-, and 7-day minimum; 1-, 3-, 7-, 30-, and 90-day maximum), as well as parameters describing other aspects of the hydrograph (low pulse count, extreme low flow frequency). Trends were evident in C2 for low flow parameters (1-, 3-, 7-, and 30-day minimum) and parameters describing other hydrograph components, including peak flows (low pulse count, extreme low peak, extreme low flow frequency, high flow peak). Full results of the single-period analysis are given in Appendix B.

The parameters that were significant for at least one watershed were used in the two-period analysis (Tables 3 and 4). The means and coefficients of variation (CV) indicate that minimum flows in U1 increased after urbanization (i.e., more flow during dry periods). Maximum flows also decreased, and CV for low and high flows decreased, indicating a less flashy flow regime for the watershed after urbanization. C2 exhibited the opposite changes in low flows with lower values and higher CV after urbanization. Other aspects of the hydrograph that changed significantly in watershed C2 suggested a flashier regime after the period of urbanization in U2: extreme low flow peaks decreased, extreme low flow frequency increased, and high flow peaks increased.

**Table 3.** Results of two-period Indicators of Hydrologic Alteration (IHA) analysis with positive change in mean values indicating an increase after urbanization for watersheds in Alabama. Coefficients of variation (CV) are given for the before (Pre) and after (Post) urbanization periods. Footnotes indicate the significance level of the trend from one period analysis. Only parameters with a significant trend for at least one watershed are presented.

| IHA Parameter | Watershed U1 | | | | Watershed C1 | | | |
|:---:|:---:|:---:|:---:|:---:|:---:|:---:|:---:|:---:|
| | Pre Mean | Post Mean | Pre CV | Post CV | Pre Mean | Post Mean | Pre CV | Post CV |
| 1-day minimum (m$^3$/s) | 61.9 | 68.8 [2] | 27.3% | 20.0% | 15.4 | 15.7 | 26.7% | 42.2% |
| 3-day minimum (m$^3$/s) | 65.7 | 74.0 [2] | 25.8% | 16.1% | 15.7 | 16.2 | 26.9% | 42.0% |
| 7-day minimum (m$^3$/s) | 70.1 | 78.3 [2] | 23.1% | 16.0% | 16.6 | 16.8 | 30.2% | 41.4% |
| 30-day minimum (m$^3$/s) | 88.3 | 96.5 | 22.5% | 18.8% | 21.0 | 21.1 | 29.1% | 48.4% |
| 1-day maximum (m$^3$/s) | 5981 | 4513 [1] | 87.2% | 43.9% | 4987 | 5450 | 41.5% | 43.1% |
| 3-day maximum (m$^3$/s) | 3545 | 2647 [1] | 90.1% | 41.7% | 3330 | 3665 | 41.1% | 46.4% |
| 7-day maximum (m$^3$/s) | 2121 | 1630 [1] | 82.6% | 34.0% | 2017 | 2090 | 39.2% | 42.9% |
| 30-day maximum (m$^3$/s) | 971 | 817 [1] | 53.1% | 29.4% | 923 | 873 | 43.1% | 34.8% |
| 90-day maximum (m$^3$/s) | 649 | 564 [1] | 43.6% | 22.0% | 620 | 560 | 39.7% | 30.6% |
| Extreme low peak (m$^3$/s) | 69.9 | 72.7 | 9.6% | 6.6% | 17.2 | 15.8 | 28.9% | 25.9% |
| Extreme low frequency (year$^{-1}$) | 7.35 | 4.68 [3] | 77.2% | 86.8% | 4.35 | 3.14 | 77.0% | 52.5% |
| High flow peak (m$^3$/s) | 964 | 861 | 23.2% | 24.2% | 1033 | 957 | 25.6% | 26.1% |

[1] Indicates $p < 0.05$, [2] Indicates $p < 0.025$, [3] Indicates $p < 0.01$.

**Table 4.** Results of two-period IHA analysis with positive change in mean values indicating an increase after urbanization for watersheds in Georgia. Coefficients of variation (CV) are given for the before (Pre) and after (Post) urbanization periods. Footnotes indicate the significance level of the trend from one period analysis. Only parameters with a significant trend for at least one watershed are presented.

| IHA Parameter | Watershed U2 | | | | Watershed C2 | | | |
|---|---|---|---|---|---|---|---|---|
| | Pre Mean | Post Mean | Pre CV | Post CV | Pre Mean | Post Mean | Pre CV | Post CV |
| 1-day minimum (m³/s) | 28.2 | 22.6 | 62.9% | 108.2% | 8.81 | 5.12 [1] | 80.6% | 144.1% |
| 3-day minimum (m³/s) | 31.1 | 23.9 | 59.4% | 108.3% | 9.15 | 5.36 [1] | 79.5% | 140.5% |
| 7-day minimum (m³/s) | 34.4 | 27.1 | 58.2% | 112.1% | 9.93 | 5.95 [2] | 74.2% | 133.7% |
| 30-day minimum (m³/s) | 59.5 | 45.1 | 55.3% | 89.5% | 19.2 | 10.7 [2] | 54.1% | 110.5% |
| 1-day maximum (m³/s) | 4113 | 4888 | 60.1% | 109.3% | 2745 | 2173 | 65.2% | 109.4% |
| 3-day maximum (m³/s) | 3417 | 3731 | 55.7% | 86.7% | 1580 | 1363 | 64.2% | 72.4% |
| 7-day maximum (m³/s) | 2114 | 2264 | 47.9% | 78.7% | 946 | 848 | 55.2% | 66.4% |
| 30-day maximum (m³/s) | 1038 | 959 | 39.5% | 48.3% | 482 | 403 | 46.5% | 67.7% |
| 90-day maximum (m³/s) | 682 | 634 | 37.7% | 39.3% | 313 | 274 | 39.5% | 53.1% |
| Extreme low peak (m³/s) | 40.4 | 33.2 | 21.6% | 28.0% | 11.2 | 10.5 [1] | 28.9% | 25.9% |
| Extreme low frequency (year⁻¹) | 3.94 | 6.50 [4] | 81.5% | 71.8% | 2.71 | 4.77 [3] | 77.0% | 52.5% |
| High flow peak (m³/s) | 948 | 1024 | 19.1% | 19.5% | 448 | 504 [4] | 22.2% | 28.9% |

[1] Indicates $p < 0.05$, [2] Indicates $p < 0.025$, [3] Indicates $p < 0.01$, [4] Indicates $p < 0.005$.

The RVA analysis (Figure 2) shows a decrease in the frequency of minimum flows in the low RVA category and increase in the high RVA category for watershed U1, consistent with the previous result of a significant increase in mean minimum flows. Over the same period, watershed C1 showed increases in both the high and low RVA categories for minimum flows, indicating an increase in variability even though there wasn't an overall significant trend. As minimal land use and land cover change occurred in this watershed, it is most likely that these alterations are the result of an increasing frequency of climate extremes, which should have also been observed at the urbanized watershed. Watersheds U2 and C2 both experienced an increase in the frequency of low values for minimum flows. Watershed C2 also showed increases in the frequency of values in the low and high range for maximum flows, but a similar increase was not observed in the urbanized watershed.

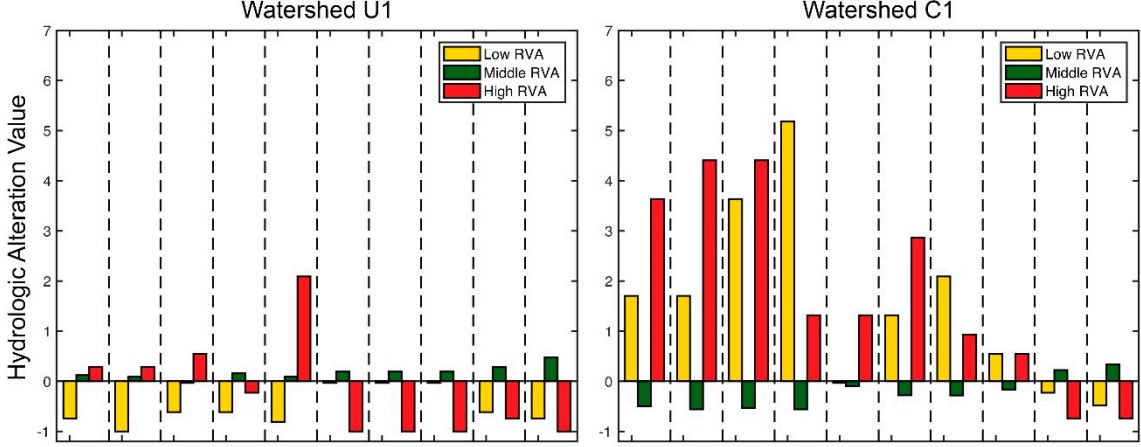

**Figure 2.** *Cont.*

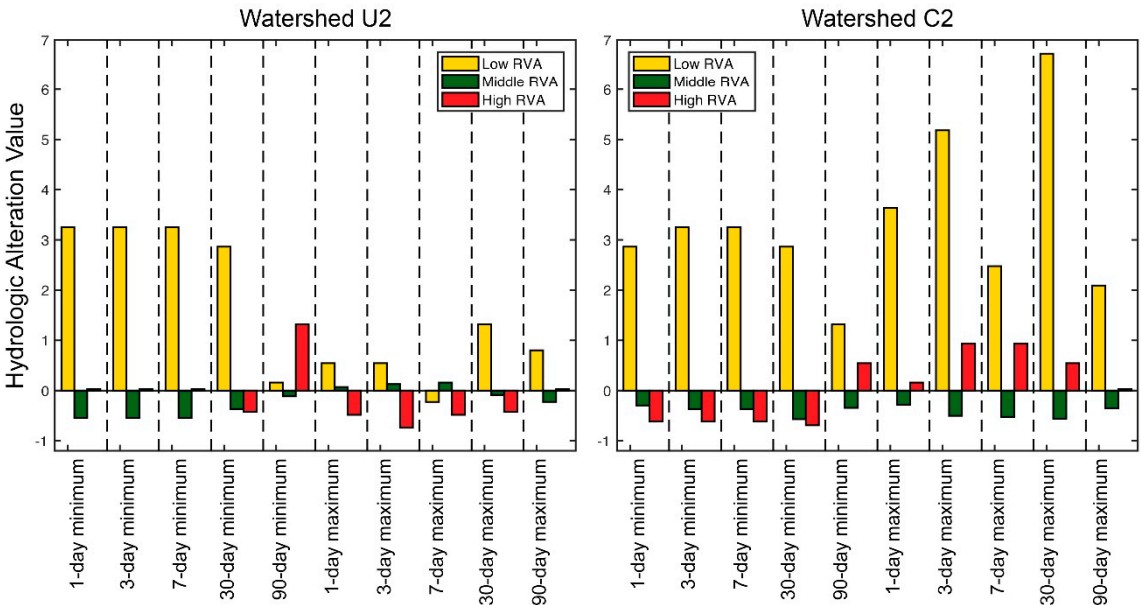

**Figure 2.** Range of variability approach (RVA) analysis by watershed. Positive hydrologic alteration values indicate a higher frequency of values in the given range of variability (low, middle, or high) after urbanization while negative values indicate a decreasing frequency.

### 3.3. Streamflow Signatures

The trends in streamflow signatures over the urbanization period for watersheds U1 and C1 were not statistically significant except for an increase in baseflow index in watershed U1 (Table 5, Figure 3). For runoff ratio, there was not a significant difference between U1 and C1 during the before-urbanization period ($p = 0.41$), but there was a significant difference for the after-urbanization period ($p < 0.01$). This suggests that the decrease in runoff ratio is significant when changes in climate are controlled for. For watershed C2, there was a significant increase in the magnitude of the slope of the FDC, indicating a flashier flow regime, but the same was not observed in watershed U2. The baseflow index decreased significantly for both watersheds U2 and C2, and thus is more likely to be related to a trend in climate or regional groundwater drawdown that has occurred in Georgia due to increased pumping and climate change [52].

**Table 5.** Mean streamflow signature values for the before- and after-urbanization periods by watershed. Footnotes indicate the *p* value of trend analysis over the study period.

| Streamflow Signatures | Watershed U1 | | Watershed C1 | | Watershed U2 | | Watershed C2 | |
|---|---|---|---|---|---|---|---|---|
| | **Before** | **After** | **Before** | **After** | **Before** | **After** | **Before** | **After** |
| Streamflow Elasticity | 0.789 | 0.791 | 0.761 | 0.813 | 0.767 | 0.777 | 0.782 | 0.794 |
| Runoff Ratio | 0.493 | 0.501 | 0.461 | 0.422 | 0.391 | 0.365 | 0.317 | 0.267 [1] |
| Slope of the FDC | −2.20 | −1.99 | −4.05 | −4.27 | −2.34 | −2.55 | −2.64 | −3.38 [4] |
| Baseflow Index | 0.634 | 0.671 [2] | 0.541 | 0.533 | 0.620 | 0.576 [3] | 0.647 | 0.597 [3] |

[1] Indicates $p < 0.05$, [2] Indicates $p < 0.025$, [3] Indicates $p < 0.01$, [4] Indicates $p < 0.005$.

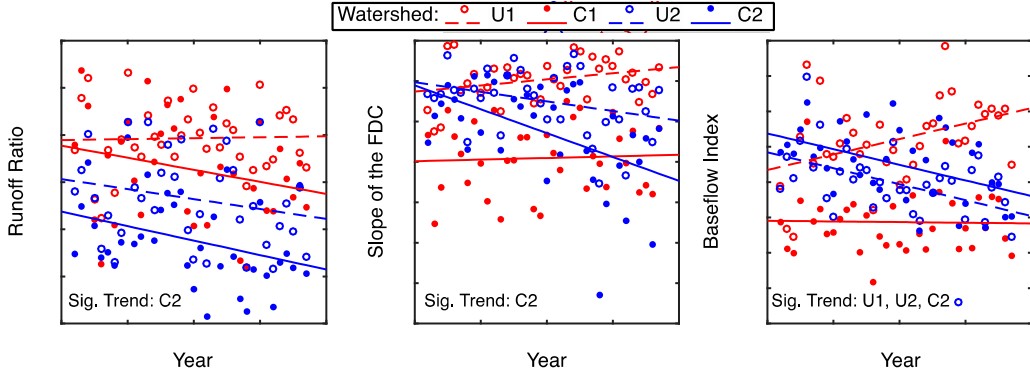

**Figure 3.** Scatterplots and trendlines for streamflow signatures for the urbanized (open symbols, dashed lines) and comparison (close symbols, solid lines) in Alabama (red) and Georgia (blue). The watersheds with significant trends are indicated in each panel. Streamflow elasticity is not shown because no significant trends were detected.

## 3.4. Hydrologic Modeling

The hydrologic model was used to simulate streamflow at ungauged locations in upstream tributaries of watershed U1 to determine if IHA and streamflow signature analysis of data for the stream gauging network were able to detect the impact of urbanization on upstream subcatchments. The area urbanized over the study periods varied substantially among the subcatchments draining to the outflow points (Figure 4). Four outflow points in the watershed were selected for further analysis (Table 6). The points are the outflows from subcatchments that represent the range of variability in urbanization change for the watershed: consistent minimal urbanization (A), transition from minimal to moderate urbanization (B), transition from minimal to heavy urbanization (C), and consistent heavy urbanization (D).

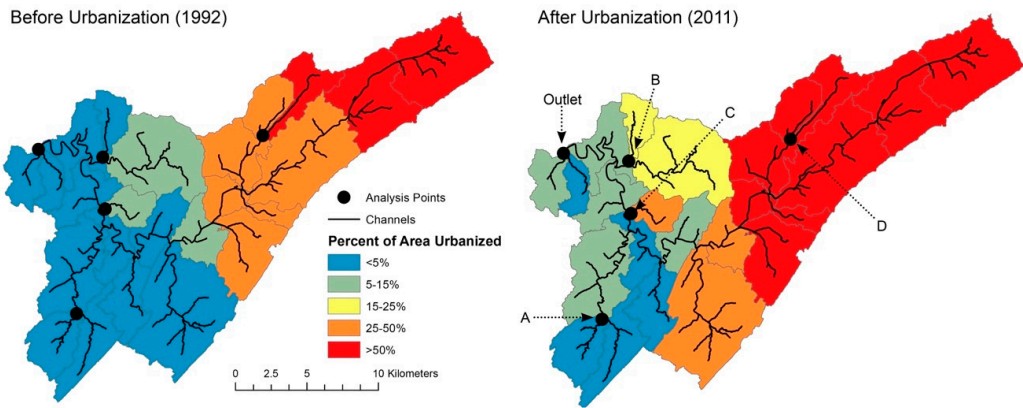

**Figure 4.** Percent of land area in each subcatchment that was classified as urbanized in 1992 and 2011. Locations of outflow points used in hydrologic modeling are shown.

**Table 6.** Change in urbanized land area for subcatchments of watershed U1 analyzed with hydrologic modeling. Initial urbanized area is from NLCD 1992. Final urbanized area is from NLCD 2011. Outlet represents the whole watershed.

| Subcatchment Outflow Point | Initial Urbanized Area (%) | Final Urbanized Area (%) |
|:---:|:---:|:---:|
| A | 1.1 | 2.9 |
| B | 3.5 | 21.6 |
| C | 9.1 | 33.7 |
| D | 52.3 | 85.3 |
| Outlet | 24.6 | 47.2 |

IHA analysis of the subcatchment outflows showed different results than for the watershed outlet (Figure 5). In subcatchments that changed from low to moderate or high levels of urbanization (B and C), much higher hydrologic alteration values were observed for the high RVA categories of minimum flows. This indicates increased flow during low flow periods. As expected, the subcatchments that had either consistently low levels of urbanization (A) or consistently high levels of urbanization (B) had hydrologic alteration values that were lower than or similar to the values observed at the watershed outlet.

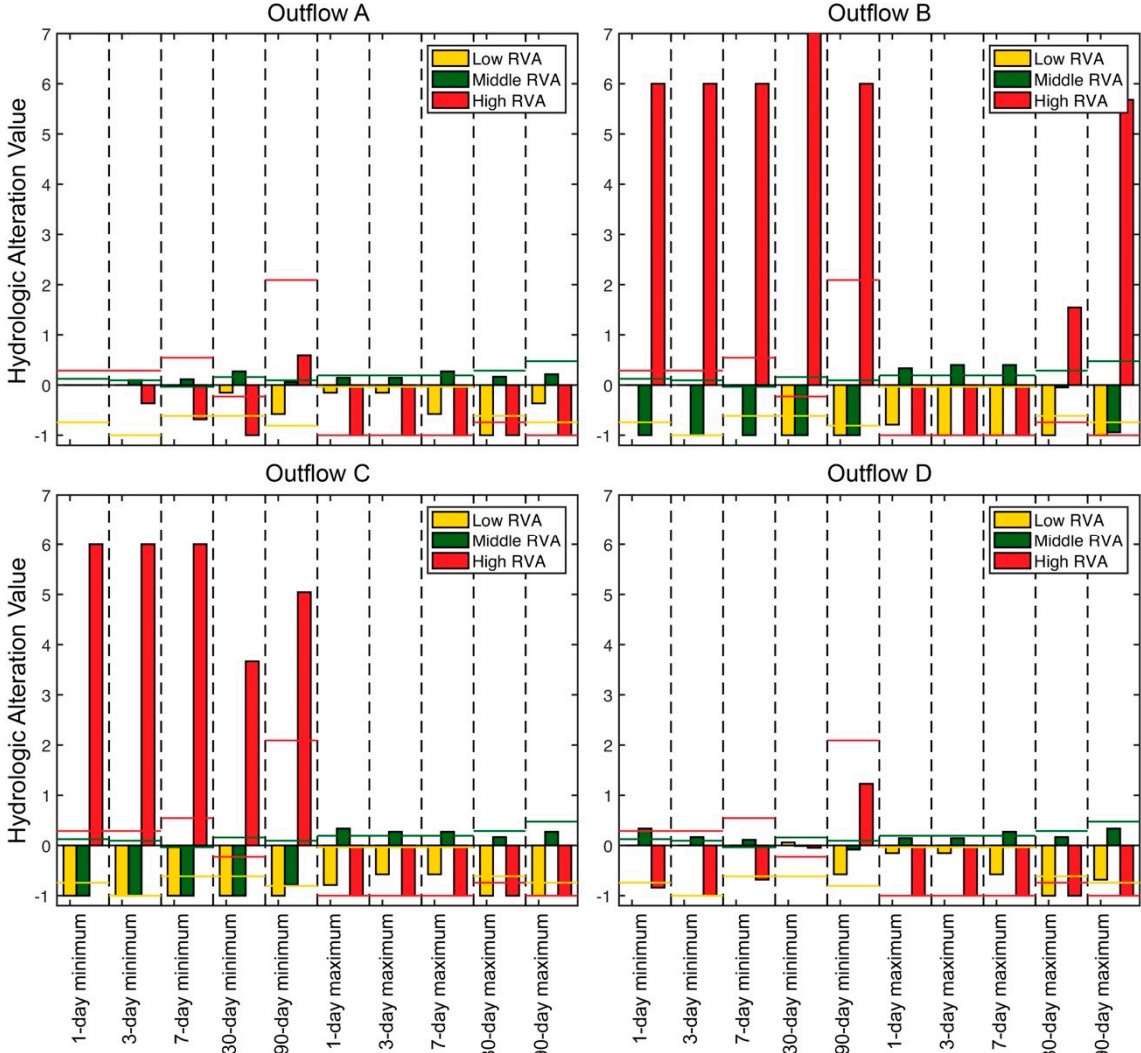

**Figure 5.** Hydrologic alteration values calculated from Hydrologic Engineering Center Hydrologic Modeling System (HEC-HMS) outputs for subcatchments spanning a range of urbanization change. Colored horizontal lines indicate the hydrologic alteration values for the watershed outlet.

Streamflow signatures showed similar results to IHA analysis (Table 7). Except for slope of the FDC at outflow A, no streamflow signatures changed significantly between the before and after urbanization periods for outflows A and D. Outflows B and C had similar results. The runoff ratio increased significantly, indicating that more precipitation was converted to runoff, while the baseflow index also increased. The magnitude of the slope of the FDC decreased, suggesting a less flashy flow regime. Overall, the magnitude of change in the streamflow signatures was greatest for outflow B.

**Table 7.** Mean streamflow signature values for the before and after urbanization period by watershed. Footnotes indicate the *p*-value of trend analysis over the study period.

| Streamflow Signatures | Outflow A | | Outflow B | | Outflow C | | Outflow D | |
|---|---|---|---|---|---|---|---|---|
| | **Before** | **After** | **Before** | **After** | **Before** | **After** | **Before** | **After** |
| Streamflow Elasticity | 0.793 | 0.783 | 0.794 | 0.792 | 0.794 | 0.785 | 0.786 | 0.788 |
| Runoff Ratio | 0.028 | 0.030 | 0.038 | 0.122 [2] | 0.319 | 0.438 [2] | 0.025 | 0.028 |
| Slope of the FDC | −4.79 | −5.21 [1] | −4.44 | −0.34 [2] | −3.91 | −2.44 [2] | −4.41 | −5.40 |
| Baseflow Index | 0.292 | 0.275 | 0.287 | 0.747 [2] | 0.472 | 0.576 [2] | 0.430 | 0.425 |

[1] Indicates $p < 0.05$, [2] Indicates $p < 0.005$.

## 4. Discussion

IHA analysis detected significant trends in both minimum and maximum flow parameters for watershed U1 (Table 3), whereas the C1 comparison watershed did not show trends over the same period. However, the trends were not what would be expected based on previous studies of urbanization impacts on hydrology. Minimum flows increased and maximum flows decreased, which is the opposite of the findings from most studies of urbanization impacts on small catchments [5,28]. though the decrease in the CV for most parameters is consistent with previous studies showing that urbanization reduces the variability of flows [4]. Over the same time period, the comparison watershed C1 had slight increases or no change in coefficient of variation, suggesting that the decreased variability in flow is indeed an effect of urbanization. The RVA analysis showed that after urbanization, the frequency of values in both the high and low RVA categories increased in watershed C1 for minimum flows and short-term maximum flows. This led to the increase in variability, but the high and low values canceled out, so there was no significant trend in the means.

The streamflow signature analysis for U1 and C1 (Table 5) gave similar results but was less sensitive in the detection of change over the study period than IHA. The baseflow index increased in U1, which is reflected in the increased minimum flow values found by IHA. While the trend in runoff ratio was not significant, comparison with watershed C1 suggested an increase in the parameter due to urbanization. Rose and Peters [8] evaluated runoff ratios for large watersheds in a comparative approach and did not find significant differences between watersheds with different levels of urbanization. The evaluation of trends in both urbanized and comparison watersheds provided additional information. Even so, the noise in hydrologic data due to climate fluctuations [53] and the relatively small differences in streamflow signatures between time periods mean that trends are difficult to detect. The change in maximum flows and decrease in variability were not detected through statistical analysis of streamflow signatures. However, there was a decreasing trend in the magnitude of the slope of the FDC ($p = 0.075$), suggesting decreased flashiness.

In the watersheds in Georgia, the comparison watershed (C2) had significant trends in several IHA parameters, but the urbanized watershed (U2) did not. C2 had significant declines in minimum flows and high flow peaks (Table 4), a pattern typically associated with urbanized watersheds [12,29]. The coefficients of variation increased in both watersheds by similar amounts. The RVA analysis also suggested that maximum flows in the low RVA category increased in C2, suggesting an overall decrease in flows in the absence of urbanization. Diem et al. [4] also analyzed the impact of urbanization in watershed U2 (Sweetwater Creek), as well as other large watersheds using trend analysis of hydrologic parameters. They found a significantly increasing trend in streamflow in several other urbanizing watersheds after normalizing by total annual precipitation totals, but did not detect a trend in watershed U2. Characteristics of rainfall other than total annual amount, such as intensity and seasonality of storms, influence runoff amounts and may need to be considered when analyzing streamflow trends [54]. Comparison with watershed C2, which would account for these factors, suggests an increasing trend in streamflow in watershed U2.

Streamflow signature analysis (Table 5) indicated a decreased baseflow index for both watersheds U2 and C2. Diem et al. [4] found a significant increase in flashiness index ($p < 0.05$), the absolute values of day-to-day changes in mean daily flow divided by the total streamflow for the year [55], for watershed U2. Slope of the FDC, which is an indicator of flashiness independent of watershed area, decreased in magnitude by 0.21 indicating increased flashiness ($p = 0.058$). The difference between the two studies suggests that the area-independent streamflow signature was less sensitive in detecting change. The magnitude of the slope of the FDC decreased by more in C2 (0.74) than in U2 and had a more strongly significant trend ($p = 0.0021$). Thus, the increase in flashiness may have been due to climate variability rather than urbanization.

Hydrologic model results suggested that hydrologic alteration was substantially greater in the subcatchments of watershed U1 that transitioned from low to high or moderate urbanization (subcatchments of outlets B and C), but the effects were mostly the opposite of what would be expected with urbanization, with both IHA analysis (Figure 5) and streamflow signatures (Table 7) indicating increased minimum flows and reduced flashiness. The runoff ratio increased in both urbanizing subcatchments, which is consistent with the expected effects of increased impervious cover [12] due to reduced infiltration. However, the increased runoff came not in larger flood peaks as would be expected, but in higher minimum flows and increased baseflow. This could represent the controlled drainage of stormwater from detention structures [30] or discharge from wastewater treatment [4]. Watershed U1 does include the Valley Creek Wastewater Treatment Plant, which may contribute to these dynamics.

As hydrologists move towards continental-scale data analysis studies, characterizing the impact of urbanization presents unique challenges that are demonstrated here. First, parameters such as streamflow signatures that are designed to be independent of watershed area for use in large-scale comparative studies appear to be less sensitive to the impacts of urbanization than other metrics, such as those used in IHA analysis. While the lack of a statistically significant result does not prove the null hypothesis, it is often the case in practice that conclusions are drawn in this manner [56]. Therefore, the lack of a trend in a low-sensitivity parameter may lead to the conclusion of no urbanization impact. Additionally, the impact of urbanization may be a change in the variability of hydrologic parameters rather than a change in mean, as shown for watershed U1 (Table 3). Second, most studies of urbanization on small watersheds show consistent and predictable effects [5,28] while studies on large watersheds, including the present study, show a mix of effects [4,8,30]. As demonstrated by hydrologic modeling, this may be due to the heterogeneity of urbanization within a large catchment. As most gauged watersheds in the NWIS are large (median size 578 km$^2$) [26], it may be necessary to integrate other sources of information, such as prediction for ungauged basins [57], to capture the effect of urbanization in analyses of stream gauge network data. It may also be due to centralized stormwater management being upstream of gauges on large watersheds [25,30], or the impact of surface water withdrawals and wastewater discharges [4]. Comprehensive, large-scale datasets on stormwater management practices, surface water withdrawals, and wastewater discharges that could be used to control for these issues is lacking. The information can be difficult to access, even in studies focused on a small number of watersheds. Numerous attempts to obtain information on stormwater management practices for Jefferson County, Alabama, (watershed U1) during the course of the present study were unsuccessful. Finally, urbanization must be analyzed as a dynamic process occurring against background climate variability. The approach used here of performing both trend analysis and comparison across watersheds made it possible to detect changes over time, such as the increase in minimum flow rates in watershed U1, while also determining which trends were more likely to be the result of climate variability, such as the increase in baseflow index and flashiness in watershed U2.

The first research question in this study was whether streamflow signatures and IHA analysis detect urbanization. IHA detected trends of decreasing maximum flows and increasing minimum flows in watershed U1. Increasing minimum flows were suggested in watershed U2 based on the decrease in minimum flows observed over the same time period in watershed C2. The second research question

was whether the change detected was consistent across watersheds in the region. This appeared to be true, as both U1 and U2 showed evidence of increased minimum flows, though the changes were more pronounced in U1. The final research question was whether the analysis of stream gauge network data detected urbanization in upstream tributaries. The changes in IHA parameters and streamflow signatures calculated from modeled streamflow were larger in magnitude but mostly in the same direction for heavily urbanizing subcatchments and the gauge. However, IHA was more likely to detect statistically significant changes when only gauge data is analyzed.

## 5. Conclusions

IHA analysis was more effective than streamflow signatures in detecting trends in hydrologic parameters during urbanization in large gauged watersheds, though the effects detected for urbanizing watersheds were the opposite of what would be expected based on studies of smaller catchments. A combination of trend analysis with comparison of urbanizing and similar non-urbanizing watersheds was a good approach for detecting change and attributing it to urbanization. Analysis of stream gauge network data from large watersheds may underestimate the importance of urbanization in determining patterns and similarities among watersheds because (1) hydrologic parameters that are independent of watershed area show low sensitivity to urbanization effects; (2) urbanization may impact small and large watershed differently due to land cover heterogeneity and water management; and (3) urbanization is a rapidly evolving watershed characteristic in many regions. Understanding of urbanization impacts on watersheds could be improved by the development of large-scale datasets on stormwater and wastewater management practices.

**Author Contributions:** Conceptualization, F.C.O.; methodology, F.C.O. and R.D.M.; software, R.D.M.; validation, R.D.M.; formal analysis, R.D.M. and F.C.O.; resources, F.C.O.; data curation, R.D.M.; writing—original draft preparation, F.C.O. and R.D.M.; writing—review and editing, F.C.O.; visualization, F.C.O. and R.D.M.; supervision, F.C.O.; project administration, F.C.O.; revision, F.C.O.

**Funding:** This research received no external funding.

**Acknowledgments:** The authors thank Xing Fang and Christopher Anderson for helpful feedback on the research presented.

**Conflicts of Interest:** The authors declare no conflict of interest.

## Appendix A. Hydrologic Model Development, Calibration, and Validation

*Appendix A.1. Curve Number Grid Development*

A gridded dataset of SCS curve numbers was developed as a hydrologic model input. NLCD land cover data was reclassified to land cover types given in the SCS TR-55 manual (Table A1). The union tool in ArcMap was used to merge the SSURGO soils map of hydrologic soil groups and the reclassified NLCD gridded land cover data into one soil and land cover polygon file. Curve numbers based on land cover and soil group (Table A2), the soil and land cover polygons, and DEM were used as inputs to the Create CN Grid tool in the GeoHMS toolbox in ArcMap to create a curve number grid.

**Table A1.** Reclassification of NLCD land cover types for curve number assignment.

| NLCD Classification | | TR-55 Re-Classification |
|---|---|---|
| **Number** | **Description** | |
| 11 | Open water | |
| 90 | Woody wetlands | Water |
| 95 | Emergent herbaceous wetlands | |
| 21 | Developed, open space | |
| 22 | Developed, low intensity | |
| 23 | Developed, medium intensity | Developed |
| 24 | Developed, high intensity | |

**Table A1.** *Cont.*

| NLCD Classification | | TR-55 Re-Classification |
|---|---|---|
| **Number** | **Description** | |
| 41 | Deciduous forest | |
| 42 | Evergreen forests | Woods |
| 43 | Mixed forest | |
| 31 | Barren land | |
| 52 | Shrub/scrub | |
| 71 | Grassland/herbaceous | Crops |
| 81 | Pasture/hay | |
| 82 | Cultivated crops | |

**Table A2.** Curve numbers used for reclassified land covers.

| Description | Hydrologic Soil Group | | | |
|---|---|---|---|---|
| | **A** | **B** | **C** | **D** |
| Water | 100 | 100 | 100 | 100 |
| Developed | 57 | 72 | 81 | 86 |
| Woods | 30 | 58 | 71 | 78 |
| Agricultural | 67 | 77 | 83 | 87 |

*Appendix A.2. Model Calibration and Validation*

The HEC-HMS model was calibrated through the optimization run manager, where parameters were varied across each subbasin. The sum of squared residuals (SSR) was used as the objective function because it gives more weight to large errors than to small errors [50]. The calibration period was water years 1981 to 1985 for the before-urbanization model and 1997 to 2000 for the after-urbanization period. The validation periods were water years 1989 to 1993 for the before-urbanization model and 2011 to 2014 for the after-urbanization period. The univariate gradient method was used for the parameter search method. The Muskingum *K* and *X* values, Initial Baseflow, Baseflow Recession Constant, Baseflow Ratio to Peak, and Initial Abstraction were calibrated to optimize agreement between observed and simulated flow, volume, and time to peak.

The performance of the calibrated model was evaluated by visually inspecting hydrographs (Figure A1) and using goodness of fit statistics for watershed simulations [58,59]. The Percent Error in Volume (PEV), Percent Error in Peak Flow (PEPF), Correlation Coefficient ($R^2$), Nash-Sutcliffe model Efficiency (NSE), and Root Mean Squared Error (RMSE) Standard Deviation Ration (RSR) were calculated and compared to the performance ratings outlined by [59]. Goodness of fit statistics are summarized in Table A3.

**Table A3.** Goodness of fit statistics for models calibrated and validated for the before urbanization (BU) period and after urbanization (AU) period. Footnotes indicate performance ratings [59].

| Header | PEV (%) | PEPF (%) | $R^2$ | NSE | RMSE-RSR |
|---|---|---|---|---|---|
| BU Calibration | 14.7 [2] | 11.7 [1] | 0.74 [2] | 0.70 [2] | 0.55 [2] |
| BU Validation | 22.8 [3] | 21.3 [2] | 0.65 [2] | 0.56 [3] | 0.66 [2] |
| AU Calibration | 4.3 [1] | 31.3 [3] | 0.74 [2] | 0.71 [2] | 0.54 [2] |
| AU Validation | 1.9 [1] | 7.9 [1] | 0.67 [2] | 0.51 [3] | 0.69 [3] |

[1] Very Good, [2] Good, [3] Satisfactory, [4] Unsatisfactory.

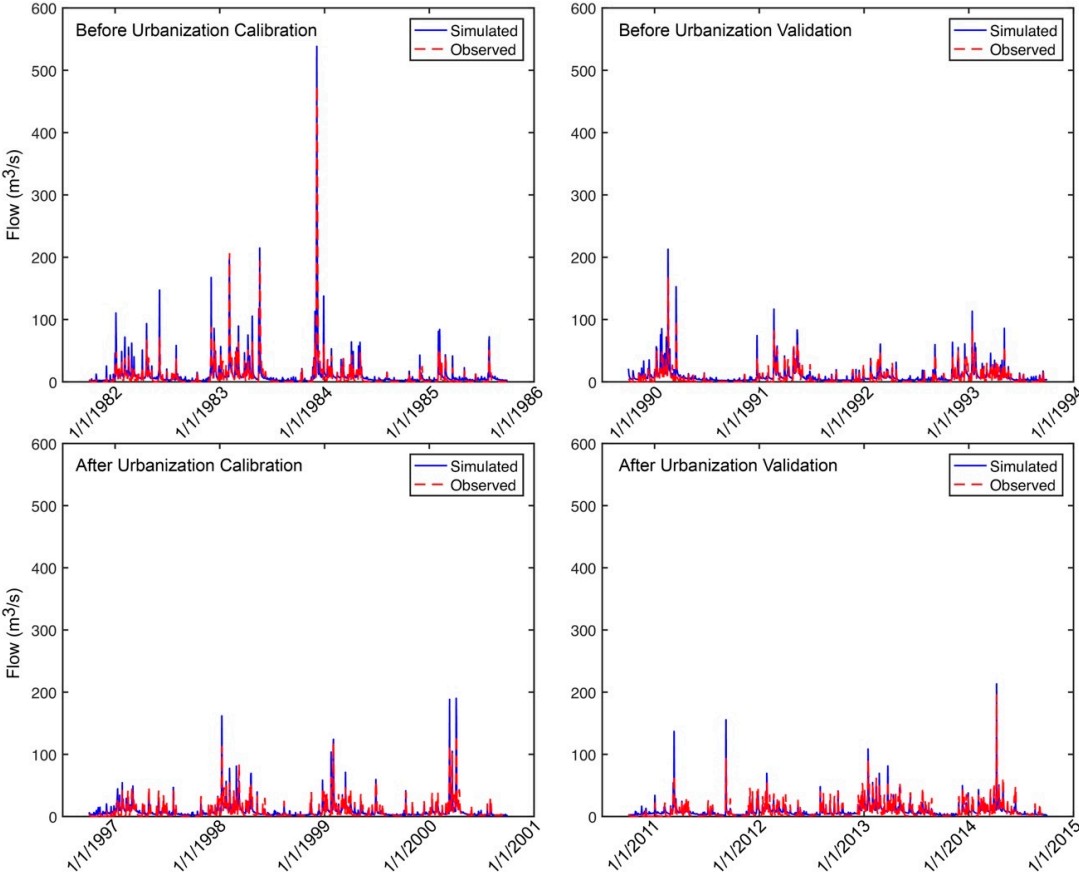

**Figure A1.** Hydrograph comparison of observed flows and flows simulated by HEC-HMS over the calibration and validation periods for the before-urbanization model and the after-urbanization model.

## Appendix B. IHA Single-Period Analysis Statistics

**Table A4.** Slope, *p*-value, and $R^2$ value for IHA single period trend analysis for watersheds U1 and C1. Positive slope values indicate an increase in the parameter.

| IHA Parameter | Watershed U1 | | | Watershed C1 | | |
|---|---|---|---|---|---|---|
| | Slope | *p* | $R^2$ | Slope | *p* | $R^2$ |
| 1-day minimum | 0.53 | 0.025 | 0.16 | −0.01 | 0.5 | 0.00 |
| 3-day minimum | 0.49 | 0.025 | 0.14 | −0.01 | 0.5 | 0.00 |
| 7-day minimum | 0.46 | 0.025 | 0.13 | −0.02 | 0.5 | 0.00 |
| 30-day minimum | 0.33 | 0.25 | 0.04 | −0.05 | 0.5 | 0.00 |
| 90-day minimum | 0.78 | 0.25 | 0.04 | −0.03 | 0.5 | 0.00 |
| 1-day maximum | −104.8 | 0.05 | 0.10 | −14.57 | 0.5 | 0.01 |
| 3-day maximum | −63.96 | 0.05 | 0.10 | −7.00 | 0.5 | 0.00 |
| 7-day maximum | −35.64 | 0.05 | 0.11 | −7.79 | 0.5 | 0.01 |
| 30-day maximum | −11.24 | 0.05 | 0.11 | −7.54 | 0.25 | 0.06 |
| 90-day maximum | −6.20 | 0.05 | 0.11 | −4.56 | 0.25 | 0.06 |
| Number of zero days | 0.00 | 0.5 | 0.00 | 0.00 | 0.5 | 0.00 |
| Base flow index | 0.00 | 0.5 | 0.03 | 0.00 | 0.5 | 0.00 |
| Date of minimum | 0.86 | 0.25 | 0.06 | −0.18 | 0.5 | 0.01 |
| Date of maximum | 1.06 | 0.5 | 0.01 | 0.64 | 0.5 | 0.00 |
| Low pulse count | −0.21 | 0.05 | 0.12 | 0.02 | 0.5 | 0.01 |
| Low pulse duration | 0.00 | 0.5 | 0.00 | −0.06 | 0.5 | 0.01 |
| High pulse count | −0.10 | 0.25 | 0.05 | −0.01 | 0.5 | 0.00 |
| High pulse duration | −0.01 | 0.25 | 0.05 | −0.02 | 0.025 | 0.14 |

**Table A4.** *Cont.*

| IHA Parameter | Watershed U1 | | | Watershed C1 | | |
|---|---|---|---|---|---|---|
| | Slope | *p* | R$^2$ | Slope | *p* | R$^2$ |
| Rise rate | −3.52 | 0.025 | 0.15 | −2.31 | 0.25 | 0.05 |
| Fall rate | 1.71 | 0.01 | 0.18 | 0.99 | 0.1 | 0.08 |
| Number of reversals | 0.24 | 0.1 | 0.08 | 0.36 | 0.05 | 0.10 |
| Extreme low peak | 0.18 | 0.1 | 0.09 | −0.06 | 0.25 | 0.06 |
| Extreme low duration | 0.00 | 0.5 | 0.00 | 0.48 | 0.1 | 0.10 |
| Extreme low timing | 1.14 | 0.25 | 0.06 | 0.06 | 0.5 | 0.00 |
| Extreme low freq. | −0.20 | 0.01 | 0.18 | −0.03 | 0.5 | 0.01 |
| High flow peak | −4.35 | 0.25 | 0.05 | −4.35 | 0.5 | 0.03 |
| High flow duration | 0.04 | 0.1 | 0.07 | 0.02 | 0.5 | 0.01 |
| High flow timing | 0.39 | 0.5 | 0.00 | 0.92 | 0.5 | 0.02 |
| High flow frequency | −0.03 | 0.5 | 0.00 | 0.05 | 0.5 | 0.03 |
| High flow rise rate | −3.38 | 0.1 | 0.07 | −2.18 | 0.5 | 0.03 |
| High flow fall rate | 2.25 | 0.005 | 0.19 | 0.19 | 0.5 | 0.00 |
| Small Flood peak | −17.11 | 0.5 | 0.06 | −6.63 | 0.5 | 0.01 |
| Small Flood duration | −0.09 | 0.5 | 0.01 | −0.09 | 0.5 | 0.01 |
| Small Flood timing | 0.84 | 0.5 | 0.01 | 0.11 | 0.5 | 0.00 |
| Small Flood freq. | 0.00 | 0.5 | 0.00 | 0.00 | 0.5 | 0.00 |
| Small Flood rise rate | −52.43 | 0.05 | 0.23 | 4.41 | 0.5 | 0.00 |
| Small Flood fallrate | 5.02 | 0.5 | 0.03 | 6.19 | 0.5 | 0.06 |
| Large flood peak | −386.5 | 0.1 | 0.67 | −6.28 | 0.5 | 0.00 |
| Large flood duration | −0.24 | 0.5 | 0.38 | −0.48 | 0.5 | 0.27 |
| Large flood timing | −6.82 | 0.5 | 0.28 | 8.34 | 0.1 | 0.73 |
| Large flood freq. | −0.01 | 0.1 | 0.07 | −0.01 | 0.5 | 0.02 |
| Large flood rise rate | 149.60 | 0.5 | 0.27 | 98.56 | 0.1 | 0.75 |
| Large flood fall rate | 31.46 | 0.1 | 0.75 | −21.01 | 0.5 | 0.15 |

**Table A5.** Slope, *p*-value, and R$^2$ value for IHA single period trend analysis for watersheds U2 and C2. Positive slope values indicate an increase in the parameter.

| IHA Parameter | Watershed U2 | | | Watershed C2 | | |
|---|---|---|---|---|---|---|
| | Slope | *p* | R$^2$ | Slope | *p* | R$^2$ |
| 1-day minimum | −0.48 | 0.25 | 0.06 | −0.23 | 0.05 | 0.13 |
| 3-day minimum | −0.53 | 0.25 | 0.07 | −0.24 | 0.05 | 0.13 |
| 7-day minimum | −0.57 | 0.25 | 0.06 | −0.25 | 0.025 | 0.13 |
| 30-day minimum | −0.75 | 0.25 | 0.05 | −0.46 | 0.005 | 0.20 |
| 90-day minimum | −1.48 | 0.25 | 0.04 | −0.94 | 0.1 | 0.09 |
| 1-day maximum | 25.25 | 0.5 | 0.00 | −36.34 | 0.25 | 0.06 |
| 3-day maximum | 7.03 | 0.5 | 0.00 | −14.79 | 0.25 | 0.04 |
| 7-day maximum | 0.34 | 0.5 | 0.00 | −6.66 | 0.5 | 0.02 |
| 30-day maximum | −5.90 | 0.5 | 0.02 | −3.85 | 0.25 | 0.05 |
| 90-day maximum | −3.78 | 0.5 | 0.03 | −2.18 | 0.5 | 0.03 |
| Number of zero days | 0.00 | 0.5 | 0.00 | 0.20 | 0.1 | 0.08 |
| Base flow index | 0.00 | 0.1 | 0.10 | 0.00 | 0.05 | 0.13 |
| Date of minimum | 0.41 | 0.5 | 0.03 | −0.18 | 0.5 | 0.01 |
| Date of maximum | 1.95 | 0.25 | 0.04 | 0.88 | 0.5 | 0.01 |
| Low pulse count | 0.14 | 0.025 | 0.15 | 0.10 | 0.025 | 0.13 |
| Low pulse duration | 0.03 | 0.5 | 0.00 | 0.23 | 0.1 | 0.08 |
| High pulse count | 0.02 | 0.5 | 0.00 | −0.04 | 0.5 | 0.01 |
| High pulse duration | −0.02 | 0.25 | 0.07 | −0.01 | 0.5 | 0.03 |
| Rise rate | 0.17 | 0.5 | 0.00 | −0.47 | 0.5 | 0.01 |
| Fall rate | 0.03 | 0.5 | 0.00 | 0.30 | 0.5 | 0.02 |
| Number of reversals | 0.19 | 0.1 | 0.09 | −0.02 | 0.5 | 0.00 |
| Extreme low peak | −0.15 | 0.25 | 0.07 | −0.08 | 0.05 | 0.16 |

**Table A5.** *Cont.*

| IHA Parameter | Watershed U2 | | | Watershed C2 | | |
|---|---|---|---|---|---|---|
| | Slope | *p* | $R^2$ | Slope | *p* | $R^2$ |
| Extreme low duration | −0.05 | 0.5 | 0.01 | 0.94 | 0.1 | 0.11 |
| Extreme low timing | −0.33 | 0.5 | 0.02 | 0.33 | 0.5 | 0.01 |
| Extreme low freq. | 0.17 | 0.005 | 0.23 | 0.10 | 0.01 | 0.17 |
| High flow peak | 1.70 | 0.5 | 0.01 | 4.97 | 0.005 | 0.25 |
| High flow duration | −0.05 | 0.25 | 0.06 | 0.00 | 0.5 | 0.00 |
| High flow timing | −1.71 | 0.25 | 0.04 | −0.39 | 0.5 | 0.01 |
| High flow frequency | −0.01 | 0.5 | 0.00 | −0.06 | 0.5 | 0.02 |
| High flow rise rate | 4.02 | 0.005 | 0.20 | 2.80 | 0.001 | 0.34 |
| High flow fall rate | −1.76 | 0.025 | 0.15 | −1.29 | 0.001 | 0.34 |
| Small Flood peak | −32.84 | 0.1 | 0.17 | 21.68 | 0.1 | 0.16 |
| Small Flood duration | −0.53 | 0.25 | 0.09 | −0.02 | 0.5 | 0.00 |
| Small Flood timing | 0.34 | 0.5 | 0.00 | 1.92 | 0.5 | 0.04 |
| Small Flood freq. | 0.01 | 0.5 | 0.01 | −0.02 | 0.25 | 0.04 |
| Small Flood rise rate | 8.46 | 0.5 | 0.04 | 6.42 | 0.5 | 0.03 |
| Small Flood fallrate | −8.45 | 0.5 | 0.07 | −2.53 | 0.5 | 0.05 |
| Large flood peak | 426.70 | 0.5 | 0.33 | −46.80 | 0.5 | 0.17 |
| Large flood duration | −0.98 | 0.5 | 0.24 | −0.66 | 0.5 | 0.04 |
| Large flood timing | 8.21 | 0.001 | 0.97 | 0.35 | 0.5 | 0.01 |
| Large flood freq. | 0.00 | 0.5 | 0.00 | 0.00 | 0.5 | 0.03 |
| Large flood rise rate | 188.40 | 0.5 | 0.35 | 10.98 | 0.5 | 0.13 |
| Large flood fall rate | −58.95 | 0.5 | 0.39 | −0.63 | 0.5 | 0.06 |

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
