# Peer review of "Assessment of Hydrologic Alteration Metrics for Detecting Urbanization Impacts"

_water, doi:10.3390/w11051017_

Round 1

Reviewer 1 Report

The study discusses the ability of streamflow signatures (e.g., runoff ratio, baseflow index) and indicators of hydrologic alteration (IHA) to detect the impacts of urbanization in large, mixed land use watersheds in the southeastern US. Results suggested that IHA can better detect streamflow changes in urban watersheds than streamflow signatures. IHA indicated increased minimum flow and baseflows and declines in flashiness in one of the urbanizing watersheds, but not the other. This finding was counter the authors’ expectation.

The results section could benefit from some additional clarification of what constitutes a significant trend for the IHA and streamflow signature and highlight mean value pre and post for each streamflow metric along the change in mean and CV reported in Tables 2 and 3. The HEC-HMS modeling component of the study is interesting but limited space is devoted to displaying and describing the streamflow metric comparisons between the four sub-catchments.

The discussion section is nicely written and makes some great points about the limitations of current statistical approaches and methodologies to detect the impacts of urban development on streamflow. Lines 387-413 provide a very nice discussion of some of these challenges. This study also demonstrates the importance of incorporating climate by using control watersheds to more accurately interpret changes in large mixed watersheds. Be sure to tie these discussion points back to the original research questions to bring this full circle.

Major Comments

Line 182: Why was the runoff ratio calculated as the mean of mean daily discharge and the mean of precipitation, rather than the sum of streamflow volume per area (yield) divided by the sum of precipitation. Using the mean rather than the sum might be part of the reason why this variable was not significant.

Lines 230-237: It would be helpful to express these land cover change stats as a percentage of the watershed area. Consider adding these to Table 1 so its easy for the reader to find. Were there any land use changes in the control watersheds? Line 252 indicates that C2 also experienced urbanization. Land cover change stats should also be provided for C1 and C2.

Lines 239-245: Consider adding a table with the results of the single period trend analyzes to the appendix. Or at a minimum report the significant level of the trends (R2 > X, p value, confidence intervals, etc.). It would also be interesting to see scatter plots of some of the trends in the appendix (e.g., one plot of 1-day max vs year with colored points for each watershed). Were the values for these streamflow metrics similar across regions? Some text should be added to the results to answer research question 2.

Lines 281-290: It is unclear why Table 3 is referred to in this paragraph about streamflow signatures. Why isn’t Table 4 referenced in this section? The captions of Table 4 and Table 6 should be rephrased to state if the table is reporting the trend results or the actual measurement of each streamflow signature.

Minor Comments:

Ling 40: Baseflow can increase or decrease in urban watersheds. Consider adding some discussion of factors contributing to variability in baseflow response after urbanization. The recent review by Bhaskar et al. 2016 provides a nice summary. Bhaskar et al. 2016. Will it rise or will it fall? Freshwater Science.

Line 66: Provide a range in watershed size to indicate the magnitude of what is considered a large basin.

Line 111: Provide additional detail for research question number 2. What types of streamflow changes were expected? Increased flashiness? Declines in baseflow?

Line 120: Indicate the magnitude of the increase in urban land cover from 1992 – 2011. Was it 10% or 50%?

Figure 1: Consider adding an aerial photo to the background of Figure 1 so the reader can see what the land use looks like in the study area. The city names are hard to read on the map. Add the locations of the weather stations mentioned in lines 140-144.

Table 1: Add statistics about current and past urban land cover to Table 1. Add the year that served as the breakpoint (before-after urbanization) to Table 1.

Line 212: What area does 20,000 cells correspond to?

Line 300: I think this should refer to Table 5, not Table 4.

Lines 366-368: Consider rephrasing this sentence to better link it to the need to incorporate rainfall intensity and seasonality in interpreting streamflow trends.

Line 379: What is (B and C) referring to? Outlets B and C?

Lines 386: Is there a wastewater treatment plant in any of the study watersheds? This seems like an important watershed characteristic to include.

Author Response

Thank you for the thorough and constructive review of our manuscript. A point-by-point response to the comments is given below.

The results section could benefit from some additional clarification of what constitutes a significant trend for the IHA and streamflow signature

The following text has been added at line 203 for IHA: "The software performs a linear regression analysis for each parameter with year as the independent variable over the period of record and calculates a p-value for the regression line to indicate the significance of the trend." The bolded text has been added at line 256 for streamflow signatures: "To determine if there was a significant trend in streamflow signatures over this period, a least-squares linear regression line was fit to the data and an F test was used to determine if the slope was significantly different from zero indicating a significant trend."

highlight mean value pre and post for each streamflow metric along the change in mean and CV reported in Tables 2 and 3.

Mean pre and post values have been added to Tables 2 and 3 (now 3 and 4).

The HEC-HMS modeling component of the study is interesting but limited space is devoted to displaying and describing the streamflow metric comparisons between the four sub-catchments.

The objective of the modeling analysis was to evaluate if streamflow metrics calculated at the stream gauge detected changes in a watershed with heterogeneous urbanization. We feel that analysis of metrics at a selection of subwatersheds that span the range of variability in urbanization change effectively and concisely addresses this objective.

Be sure to tie these discussion points back to the original research questions to bring this full circle.

The following text has been added to the end of the discussion: "The first research question in this study was whether streamflow signatures and IHA analysis detect urbanization. IHA detected trends of decreasing maximum flows and increasing minimum flows in watershed U1. Increasing minimum flows were suggested in watershed U2 based on the decrease in minimum flows observed over the same time period in watershed C2. The second research questions was whether the change detected was consistent across watersheds in the region. This appeared to be true, as both U1 and U2 showed evidence of increased minimum flows, though the changes were more pronounced in U1. The final research question was whether the analysis of stream gauge network data detected urbanization in upstream tributaries. The changes in IHA parameters and streamflow signatures calculated from modeled streamflow were larger in magnitude but mostly in the same direction for heavily urbanizing subcatchments and the gauge. However, IHA was more likely to detect statistically significant changes when only gauge data is analyzed."

Line 182: Why was the runoff ratio calculated as the mean of mean daily discharge and the mean of precipitation, rather than the sum of streamflow volume per area (yield) divided by the sum of precipitation. Using the mean rather than the sum might be part of the reason why this variable was not significant.

Daily data is used for both Q and P, so the ratio of the means is [sum(Q)/365]/[sum(P)/365]. 1/365 cancels out, so it is the same value that would be given as taking the ratio of the sums. We have changed "means" to "sum" (now at line 230) to avoid confusion. 

Lines 230-237: It would be helpful to express these land cover change stats as a percentage of the watershed area. Consider adding these to Table 1 so its easy for the reader to find. Were there any land use changes in the control watersheds? Line 252 indicates that C2 also experienced urbanization. Land cover change stats should also be provided for C1 and C2.

These stats have been added as Table 2. Line 252 is intended to state that this occurred during the period in which urbanization was occurring in U1 and U2. It has been changed (now at line 314) from "C2 suggested a flashier regime after urbanization" to "C2 suggested a flashier regime after the period of urbanization in U2" to clarify.

Lines 239-245: Consider adding a table with the results of the single period trend analyzes to the appendix. Or at a minimum report the significant level of the trends (R2 > X, p value, confidence intervals, etc.). It would also be interesting to see scatter plots of some of the trends in the appendix (e.g., one plot of 1-day max vs year with colored points for each watershed). Were the values for these streamflow metrics similar across regions? Some text should be added to the results to answer research question 2.

The single period statistics have been added as Appendix B. Given that the watersheds have different areas, plotting the IHA parameters, such as 1-day max, on the same plot presents a challenge as the magnitudes are quite different. Therefore, we added scatter plots of the streamflow signatures, which are normalized by area, as Figure 3.

Lines 281-290: It is unclear why Table 3 is referred to in this paragraph about streamflow signatures. Why isn’t Table 4 referenced in this section? The captions of Table 4 and Table 6 should be rephrased to state if the table is reporting the trend results or the actual measurement of each streamflow signature.

The table reference has been corrected (it is now Table 5). In the captions for tables 5 and 7, we changed "Streamflow signatures" to "Mean streamflow signature values" to clarify what is presented.

Ling 40: Baseflow can increase or decrease in urban watersheds. Consider adding some discussion of factors contributing to variability in baseflow response after urbanization. The recent review by Bhaskar et al. 2016 provides a nice summary. Bhaskar et al. 2016. Will it rise or will it fall? Freshwater Science.

This reference was added along with the text "the effect of urbanization on baseflow is complex and may lead to increases or decreases."

Line 66: Provide a range in watershed size to indicate the magnitude of what is considered a large basin.

Added "(approximately 50-1000 km2)" now at line 72.

Line 111: Provide additional detail for research question number 2. What types of streamflow changes were expected? Increased flashiness? Declines in baseflow?

Research question 2 has been changed to: "Is the change detected consistent across watersheds within a region and with the expected increase in flashiness associated with urbanization?"

Line 120: Indicate the magnitude of the increase in urban land cover from 1992 – 2011. Was it 10% or 50%?

This information is now included in Table 2.

Figure 1: Consider adding an aerial photo to the background of Figure 1 so the reader can see what the land use looks like in the study area. The city names are hard to read on the map. Add the locations of the weather stations mentioned in lines 140-144.

 Figure 1 has been changed to incorporate these suggestions.

Table 1: Add statistics about current and past urban land cover to Table 1. Add the year that served as the breakpoint (before-after urbanization) to Table 1.

The statistics now appear in Table 2. Breakpoint years have been added to Table 1.

Line 212: What area does 20,000 cells correspond to?

This has been changed to "20,000 elevation raster cells (approximately 2 km2)" now at line 266.

Line 300: I think this should refer to Table 5, not Table 4.

Corrected. It is now Table 6.

Lines 366-368: Consider rephrasing this sentence to better link it to the need to incorporate rainfall intensity and seasonality in interpreting streamflow trends.

This sentence has been changed to "Characteristics of rainfall other than total annual amount, such as intensity and seasonality of storms, influence runoff amounts and may need to be considered when analyzing streamflow trends." Now at line 630

Line 379: What is (B and C) referring to? Outlets B and C?

Yes. It has been changed to "subcatchments of outlets B and C" now at line 644.

Lines 386: Is there a wastewater treatment plant in any of the study watersheds? This seems like an important watershed characteristic to include

The following text has been added at line 651: "Watershed U1 does include the Valley Creek Wastewater Treatment Plant, which may contribute to these dynamics."

Reviewer 2 Report

This study used two methods to explore our ability to detect the effects of urbanization on small streams using stream gage data for their larger receiving streams.  They had two pairs of watersheds, one urbanized and one control, in the Piedmont region in the Southeastern United States. They used historic gage data to compare how well the Indicators of Hydrologic Alteration and streamflow signatures methods could detect the hydrologic changes due to urbanization in the urbanized watersheds.  They also used a HEC-HMS model and land cover data in ARC-GIS to look at subwatershed changes within one of their urbanized watersheds. They expected to find lower baseflows and higher peak flows in the small streams with urbanization.  Instead, they found mixed results, including often higher baseflows and lower peak flows.  They suggest this difference from their hypothesis could be due to the limitations of the model or possibly due to the effectiveness of stormwater management within these watersheds.  Throughout their study, the IHA appeared to be better at detecting the difference between the urbanized and control watersheds. Because some changes due to urbanization were detected, even if they were not the expected changes, they recommend the use of the IHA in future studies, while recognizing the need for more large-scale data particularly on smaller streams rather than trying to do this sort of extrapolation from the receiving waters about the effects of urbanization on the headwaters.

This is a limited study using readily available datasets and methods to attempt to examine something that would often require much more extensive instrumentation, field work, and/or long term datasets that are currently lacking to detect.  Although they had limited success in their pilot study, it still appears to be a useful exercise in exploring how we can best monitor hydrologic changes in urbanizing watersheds.

line 39-40: The effects of urbanization on baseflow are known to vary, with some studies finding increases due to leaky infrastructure.  See the review in Bhaskar et al in 2016 in Freshwater Science.

line 121/Table 1: U2 is much bigger than the other watersheds.  That seems like a real confounding effect that should be tested for or at least acknowledged and discussed.

Table 1: I suggest including stream names here.  That will make this more helpful to future studies of those streams. 

line 335-336: Again, there are studies of small streams that show increased minimum baseflow, particularly in places with aging, leaky infrastructure.

line 396: Again, I disagree that there has been observed a consistent effect of urbanization on small streams.

Author Response

Thank you for reviewing our manuscript. Point-by-point responses to your comments are given below.

line 39-40: The effects of urbanization on baseflow are known to vary, with some studies finding increases due to leaky infrastructure. See the review in Bhaskar et al in 2016 in Freshwater Science.

This reference was added along with the text "the effect of urbanization on baseflow is complex and may lead to increases or decreases."

line 121/Table 1: U2 is much bigger than the other watersheds. That seems like a real confounding effect that should be tested for or at least acknowledged and discussed.

The following text has been added, now at line 126: “Watershed U2 is somewhat larger than the other watersheds. Therefore, comparisons of the magnitude of parameters that are not normalized by area should be done cautiously when this watershed is considered. C2, despite being smaller, still offers a useful comparison for trends in parameters and normalized parameters such as streamflow signatures."

Table 1: I suggest including stream names here. That will make this more helpful to future studies of those streams. 

The stream names have been added to Table 1 and the full names of the gauging stations have been added to the main text at line 146.

line 335-336: Again, there are studies of small streams that show increased minimum baseflow, particularly in places with aging, leaky infrastructure.

line 396: Again, I disagree that there has been observed a consistent effect of urbanization on small streams.

Based on our review of the literature, it is more common for baseflow to decrease in small streams after urbanization particularly in the southeastern U.S. Line 335 (now 356) has been changed to "most studies of urbanization impacts on small catchments," and line 396 (now 418) has been changed to "most studies of urbanization on small watersheds."